# Best-Practice Aspects of Quantum-Computer Calculations: A Case Study of the Hydrogen Molecule

**DOI:** 10.3390/molecules27030597

**Published:** 2022-01-18

**Authors:** Ivana Miháliková, Martin Friák, Matej Pivoluska, Martin Plesch, Martin Saip, Mojmír Šob

**Affiliations:** 1Institute of Physics of Materials, Czech Academy of Sciences, v.v.i., Žižkova 22, CZ-616 62 Brno, Czech Republic; mihalikova@ipm.cz (I.M.); friak@ipm.cz (M.F.); mojmir@ipm.cz (M.Š.); 2Institute of Computer Science, Masaryk University, Šumavská 416, CZ-602 00 Brno, Czech Republic; pivoluskamatej@gmail.com; 3Department of Condensed Matter Physics, Faculty of Science, Masaryk University, Kotlářská 2, CZ-611 37 Brno, Czech Republic; 4Department of Chemistry, Faculty of Science, Masaryk University, Kotlářská 2, CZ-611 37 Brno, Czech Republic; saip.martin@gmail.com; 5Institute of Physics, Slovak Academy of Sciences, Dúbravská Cesta 9, SK-841 04 Bratislava, Slovakia

**Keywords:** quantum computers, hydrogen molecule, variational quantum eigensolver, circuit architecture, quantum computing, quantum chemistry, COBYLA, SPSA

## Abstract

Quantum computers are reaching one crucial milestone after another. Motivated by their progress in quantum chemistry, we performed an extensive series of simulations of quantum-computer runs that were aimed at inspecting the best-practice aspects of these calculations. In order to compare the performance of different setups, the ground-state energy of the hydrogen molecule was chosen as a benchmark for which the exact solution exists in the literature. Applying the variational quantum eigensolver (VQE) to a qubit Hamiltonian obtained by the Bravyi–Kitaev transformation, we analyzed the impact of various computational technicalities. These included (i) the choice of the optimization methods, (ii) the architecture of the quantum circuits, as well as (iii) the different types of noise when simulating real quantum processors. On these, we eventually performed a series of experimental runs as a complement to our simulations. The simultaneous perturbation stochastic approximation (SPSA) and constrained optimization by linear approximation (COBYLA) optimization methods clearly outperformed the Nelder–Mead and Powell methods. The results obtained when using the $R_{y}$ variational form were better than those obtained when the $R_{y}R_{z}$ form was used. The choice of an optimum entangling layer was sensitively interlinked with the choice of the optimization method. The circular entangling layer was found to worsen the performance of the COBYLA method, while the full-entangling layer improved it. All four optimization methods sometimes led to an energy that corresponded to an excited state rather than the ground state. We also show that a similarity analysis of measured probabilities can provide a useful insight.

## 1. Introduction

Quantum computing has recently emerged as a very promising alternative to conventional computational means. Conventional supercomputers, albeit versatile and remarkably reliable, seem to be outpaced by ever-increasing demand for computational power when developing new drugs [1], modeling nanoparticles [2], or assessing problems in materials science [3] and nuclear physics [4,5]. In contrast to the well-established conventional technologies, quantum computers are expected to provide exponentially growing computational power thanks to the their use of quantum effects [6,7], and the first indications of so-called quantum advantage/supremacy have already been demonstrated [8,9,10]. Unfortunately, the current capabilities of quantum computers are still rather limited by numerous methodological issues, a lack of suitable software tools, challenges when physically realizing quantum circuits, noise, which reduces their reliability, as well as a very low number of quantum platforms available for users.

Despite the above-mentioned challenges, the onset of quantum computers is undeniable, and quantum chemistry is one of the most active areas of their applications [11,12,13,14,15]. In particular, quantum-mechanical calculations of the properties of small molecular systems represent one of the most successful utilizations of quantum calculations [16,17,18,19]. Importantly, quantum computers are no different from their classical counterparts regarding numerous technicalities that are critical for their successful performance [20]. The architecture of quantum circuits, the optimization methods used to reach the energy minimum, and numerous computational parameters critically affect the calculations.

Our study aimed at identifying the impact of different computational setups, and we used the ground-state energy of the H${}_{2}$ molecule as a benchmark for which the exact solution exists. Such an initial testing of the setups is important whenever starting a new set of calculations, e.g., for a new molecular system or a different molecular property. Our results clearly show that extensive testing can not only reduce the use of valuable computational resources, but can be also critical for obtaining the correct minimum-energy state.

## 2. Results

Two examples of the optimization process, when the ground-state energy of the H${}_{2}$ molecule is searched, are presented in Figure 1.

The two subfigures in Figure 1 present energies as a function of the number of iterations (functional evaluations) for either the constrained optimization by linear approximation (COBYLA) method [21] or the simultaneous perturbation stochastic approximation (SPSA) [22,23,24] optimization method. The results were obtained for a four-qubit BK-transformed Hamiltonian with the circuit architecture characterized by the $R_{y}R_{z}$ variational form and the linear entangling layer of qubits. The comparison of both converging trends is interesting. On the one hand, both optimization methods converged to very similar final energies within a rather similar number of iterations, and importantly, both converged energies were very close to the exact value. On the other hand, Figure 1 also clearly shows that the iterative process itself was very different and sensitive to the optimization method used.

COBYLA optimization is a gradient-free method that provides energies decreasing by smaller amounts, but rather monotonously (we leave aside a certain level of numerical noise). In contrast to that, the SPSA method requires a series of initial iterations for the evaluation of pseudo-gradients, but once these are determined, the energy decreases more abruptly by a larger amount in a smaller number of iterations.

### 2.1. Performance of Various Optimization Methods

While the above-discussed examples are illustrative, it is hard to draw conclusions from a single simulation run for each optimization method. Therefore, below, we exhibit the results for a series of 50 calculations with a randomly selected initial set of angles defining the state of qubits. Further, we analyzed also two additional optimization methods, a classical Nelder–Mead [25] and that of Powell [26], next to COBYLA and SPSA. The predicted ground-state energies of the hydrogen molecule for 50 simulations using these four optimization methods are shown in Figure 2.

Clearly, there are rather striking differences in the performance of the analyzed methods. As far as the COBYLA method is concerned (see Figure 2a), most of the simulations resulted in energies that are quite far away from the exact one covering a range as wide as 0.3 Ha between the exact value (−1.867 Ha) and −1.65 Ha. Two calculations provided energies that are clearly very far away from the ground state of H${}_{2}$, and these energies are, in fact, quite close to those of the excited states of the hydrogen molecule. In particular, the eigenvalues of the four-qubit Hamiltonian of the H${}_{2}$ molecule, as determined using classical techniques, are as follows: −1.867,−1.262,−1.262,−1.242,−1.242,−1.242,−1.160,−1.160,
−0.881,−0.465,−0.465,−0.341,−0.341,−0.211,0.000,0.227 Ha.

The values between −1.3 Ha and −1.2 Ha in Figure 2a can correspond to excited states with the energy equal to −1.262 Ha or −1.242 Ha. While these energies of the excited states are in principle interesting and physically plausible (as a solution for some valid states of the H${}_{2}$ molecule), we should keep in mind that they represent incorrect predictions as far as our search for the ground-state energy is concerned.

The SPSA method (see Figure 2b) offers most of the energies very close to the exact value with only two computational runs ending up in the energy region corresponding to the excited states and only one simulation providing a value that is clearly erroneous. The Nelder–Mead method (see Figure 2c) clearly failed to converge to the correct value in a vast majority of the cases with only 5–8 simulations, i.e., 10–16%, providing the correct energy. While about 20% of the results between −1.2 Ha and −1.3 Ha could be assigned to the excited states, all other values are clearly erroneous.

Lastly, the Powell method (see Figure 2d) represents an intermediate case as far as the accuracy is concerned. While the spread of the computed energies of the ground state was smaller than that obtained for the COBYLA method, the energies were predicted twice more often in the region corresponding to the excited states (when compared with the COBYLA results).

### 2.2. Impact of the Variational Form

After analyzing the performance of different optimization methods, we next focused on the circuit architecture, and we started with the variational form. We compared our initial one in Figure 3 with an alternative one in Figure 4.

They differ in the number of gates. While the results in Figure 2 were obtained for the $R_{y}R_{z}$ variational form (see Figure 3), Figure 5 summarizes a corresponding set of results in the case of the $R_{y}$ variational form (that is shown above in Figure 4). The comparison of the results in Figure 2 and Figure 5 clearly shows that the use of the $R_{y}$ variational form led to more accurate results even in the case of the Nelder–Mead method. We, therefore, used the $R_{y}$ variational form in our simulations below, unless specified otherwise.

### 2.3. Influence of the Details of the Entangling Layers

Next, we focused on another aspect of the circuit architecture, the entangling layers of qubits. After performing all the above calculations with the linear entangling layers (see the light-blue cNOTs in the central part of both Figure 3 and Figure 4), we took into account also the circular and full-entangling layers visualized in Figure 6.

The SPSA method proved to be rather insensitive to changes in the entangling layer; see Figure 6a,b. In contrast to this robustness, the results obtained when using the COBYLA method clearly showed how sensitive this method is to the characteristics of the entangling layer.

While the circular entangling layer significantly increased the number of erroneous results (see Figure 7c), the full-entangling layer significantly reduced the number of energies in the excited-state region from six (see Figure 5a) to two (see Figure 7d), but three values ended in the erroneous region between the ground state and the excited ones.

### 2.4. Analysis of the Probabilities of the Basis States

Observing results corresponding to excited states rather than the ground state, we address this issue in a detail. As discussed in the text related to Figure 2 above, there were a few eigenvalues in the energy range between −1.2 Ha and −1.3 Ha. Assuming that the information about the phases was not available and we were limited to measuring only the probabilities of the basis states, below, we suggest that the computed energies were sorted according to the similarity of these measured probabilities.

Motivated by the fact that the COBYLA method seems to be more prone to providing results corresponding to the excited states, but otherwise performing quite well, we focused our analysis on this method. Moreover, in order to maximize the accuracy, below, we used the maximum value of shots, i.e., 8192, allowed by the Qiskit package.

There are $2^{l}$ basis states for an *l*-qubit Hamiltonian, and therefore, we analyzed the probabilities of the following 16 basis states: |0000〉, |0001〉, |0010〉, |0011〉, |0100〉, |0101〉, |0110〉, |0111〉, |1000〉, |1001〉, |1010〉, |1011〉, |1100〉, |1101〉, |1110〉, |1111〉. The following sets show examples of the measured probabilities:

Set A0: (22, 4, 9, 81, 126, 0, 34, 0, 1, 2, 1, 0, 29, 0, 7880, 3)/8192 for Circuit 0;

Set A1: (10, 9, 7, 23, 1220, 0, 2378, 2, 1, 11, 19, 31, 2543, 0, 1935, 3)/8192 for Circuit 1.

These result in an energy of −1.8422 Ha. The other sets are:

Set B0: (21, 2, 2, 8, 183, 0, 106, 1, 2, 0, 4, 0, 12, 0, 7839, 12)/8192 for Circuit 0;

Set B1: (0, 3, 16, 10, 1111, 2, 2026, 2, 3, 1, 7, 0, 3286, 8, 1714, 3)/8192 for Circuit 1.

These result in an energy of −1.8464 Ha. As both energies are close to the exact value of the ground state, we assumed that the sets of probabilities corresponded to the ground state and that differences between them were due to the probabilistic nature of quantum states (in the case of simulating an ideal quantum processor). When inspecting Sets A0 and B0 for Circuit 0, it seemed that the probability of the basis state |1110〉 was very close to one and nearly zero otherwise. In such a situation, it is easy to tell that the following set of measured probabilities for Circuit 0:

Set C0: (3, 2, 208, 7269, 0, 13, 28, 6, 7, 19, 2, 132, 4, 262, 7, 230)/8192 for Circuit 0, is likely related to a different eigenvalue. Indeed, the corresponding energy was equal to −1.2526 Ha, i.e., a higher-lying eigenvalue corresponding to an excited state. The situation, when the probability is very close to one for one of the basis states and nearly zero for all the others, is advantageous in the case of noise and errors containing runs because it opens the way towards identifying the probabilities that are caused by errors and noise. They can then be used in a reverse-engineering manner as parameters in noise-mitigation techniques. Ideally, we would like to have a tool that can identify Sets A0 and B0 as similar, while Set C0 as very dissimilar. Before we suggest a suitable similarity measure, it is worth discussing the probabilities related to Circuit 1 as well.

As far as Sets A1 and B1 of the probabilities for Circuit 1 are concerned, the situation is not as clear as in the case of Sets A0 and B0 for Circuit 0. In particular, there are four basis states with significant outcome probabilities, but the probabilities for the same basis states in Sets A1 and B1 differ by more than 10%. A high number of shots is then needed to determine the probabilities’ reliability. Again, it would be advantageous to have a similarity measure that identifies A1 and B1 as similar and related to the same eigenvalue.

Considering that the probabilities are sets of non-negative values, we suggest using the following two measures. First is the Jaccard–Tanimoto (J-T) index, also known as the Jaccard similarity coefficient, which is used for gauging the similarity and diversity of sample sets [27,28,29,30,31]. It was developed by P. Jaccard [32] and independently formulated again by T. Tanimoto [33]. The Jaccard–Tanimoto index of two sets X and Y is defined as the ratio of the intersection of the two sets over their union J(X,Y)=|X∩Y|/|X∪Y|. For two vectors {$x_{i}$}, {$y_{i}$} with all components non-negative ($x_{i}\geq 0$, $y_{i}\geq 0$) and the same length (i=1,...,m), it is evaluated as $J\left(\left\{x_{i}\right\},\left\{y_{i}\right\}\right)=\sum_{i}\min\left(x_{i},y_{i}\right)/\sum_{i}\max\left(x_{i},y_{i}\right).$

The second measure is the scalar product of the vectors representing the set of measured probabilities. It must be emphasized that the vectors of measured probabilities have a sum of components equal to one, but their length, in a vector sense, is in general not equal to one. The length of a probability vector is equal to one only when one of the basis states has the probability close to one and all others have the probability equal to zero. It is also the maximum length in the vector sense. The other extreme case, when the length of the vector of probabilities is lowest, is the situation when all *m* basis states have the same probability equal to 1/*m* and the length is 1/$\sqrt{m}$. Therefore, we normalized the probability vectors before we evaluated their scalar product as follows. Considering that the probabilities are the squares of the amplitudes of wave functions, we used the square roots of the measured probabilities when evaluating the similarity by the vector product. Defined in this way, the scalar product as the second similarity measure corresponds to the upper bound of the fidelity between the states. Below, we evaluated the similarities as obtained for two sets (for Circuits 0 and 1) of 500 vectors of measured probabilities.

Having the two similarity measures for our analysis, we applied them in a general manner assuming no prior knowledge of which set of probabilities is related to which eigenvalues. Our choice was motivated by the fact that future applications of quantum computers will likely be focused on systems for which data similar to those that we have at hand for the H${}_{2}$ molecule will not be available. Therefore, we determined for each vector in those two 500-member sets (for Circuits 0 and 1) the J-T/scalar-product similarities of each particular vector with all 500 vectors in the set, and then, we determined the average over these 500 similarity values. Figure 8 shows the values of both the J-T similarity index (Figure 8a,b) and scalar product (Figure 8c,d) for both Circuit 0 (Figure 8a,c) and Circuit 1 (Figure 8b,d).

Figure 8 shows that these averaged similarity values decreased with increasing energy from the ground-state value. Starting with Circuit 0, as many as 450 from the 500 runs resulted in a set of probabilities that were similar to the situation when the probability was close to one for the |1110〉 basis state and nearly zero otherwise. The corresponding energies covered the range from the exact value of −1.867 Ha up to −1.7 Ha. The J-T similarity index values (see Figure 8a) show a weakly decreasing trend with increasing energy. In contrast, the averaged scalar products of all of these 450 probability vectors were very close to the same value 0.95; see the results in Figure 8c. As they represent an average over 450 very similar states (when the similarity values are close to one), but about 10% of very different states (these contribute to the average by the similarity values close to zero), the average values are not equal to one, but only to about 0.95.

As another extreme, about 5% of the vectors of probabilities had very low averaged similarity values (close to zero), and the corresponding energies belonged to the range between −1.3 Ha and −1.2 Ha. These we identified as excited states, and their vectors of probabilities were very different. Interestingly, while the J-T similarity index assigns to these cases’ values that are clearly close to zero without any exception, the scalar product in one case is a value close to 0.5. The remaining 5% of probability vectors were characterized by energies and similarity index values that were in between the region close to the ground state, on the one hand, and that of the excited states, on the other. We believe that these essentially erroneous calculations should be dropped.

Circuit 1 (see Figure 8b,d) turned out to be more complicated. The ground state possessed the probability vector in the form of a superposition of four basis states (|0100〉, |0110〉, |1100〉, and |1110〉) with rather similar probabilities (they sometimes differed by less than 1/(2${}^{l}$, where *l* is the number of qubits). As far as the J-T similarity index is concerned, it performed qualitatively similarly (see Figure 8b) as in the case of Circuit 0 (see Figure 8a), but some of the states that we identified as excited showed significant non-zero values (up to 0.4). These features were even more pronounced in the case of the scalar product (see Figure 8d).

### 2.5. Simulations of Noisy Runs

Our analysis so far has been based on simulations of ideal quantum processors that do not exhibit any noise. While these simulations currently represent a major part of published quantum-computing results, our ultimate goal is to employ real quantum processors. The current ones based on superconductor units are, unfortunately, quite noisy, partly due to their quantum nature [34], but mostly due to unresolved issues related to the technical complexity of physical realizations of quantum processors; for details, see, e.g., [35]. Consequently, there is a tremendous effort focused on various error-mitigation methods [36,37,38,39,40,41].

Motivated by the above-mentioned facts, we extended our theoretical analysis to simulations that included noise. Its characteristic parameters are frequently published by IBM for its quantum processors. Importantly, it is possible to even switch on/off different types of noise in simulations, in particular the noise related to gate errors, readout errors, and the combination thereof. Our results are presented in Figure 9 for both the SPSA and COBYLA optimization methods in comparison with noise-free ideal simulator values.

When considering all possible errors (see the red-colored data in Figure 9), the use of both optimization methods resulted in a similar shift by about 0.07 Ha to higher energies. Simulations including solely gate errors or readout errors then showed that the latter errors were responsible for the dominant contribution to the energy shift. In contrast, the gate errors resulted in a lesser part of the energy shift. The comparison of Figure 9a,b indicated that the results related to the COBYLA method covered wider ranges when compared with the SPSA results, but the median values were similar.

It is worth noting that some energies in Figure 9 are lower (more negative) than the exact value. This seemingly contradicts the application of the variational principle, but the reason was, in fact, the impact of noise again. In particular, as the probabilities were noisy and the coefficients in the Hamiltonian (Equation 1) had both positive and negative values, a noise-related re-distribution of probabilities from positive coefficients to negative ones may result in energies that are even lower than the exact value.

### 2.6. Experiments on Real Quantum Processors

After running the noise-containing simulations, our final step consisted of experiments using physical realizations of quantum processors. As these are resource-intensive, we limited ourselves only to the two-qubit Hamiltonian. Figure 10 exemplifies the similarity of the results for the four-qubit and two-qubit Hamiltonians in the case of ideal noise-free simulations.

After confirming the similarity of the results when using both two-qubit and four-qubit Hamiltonians, we present the values obtained during ten experiments consisting of iterative runs employing two IBM quantum processors in Figure 11. By inspecting the resulting energies, it is interesting to note that while all experimental runs behaved quite similarly as a function of the number of functional evaluations and they all converged to a similar energy, none of them converged to the exact value of the ground-state energy of the H${}_{2}$ molecule. Instead, there was a small shift to a slightly higher energy. The origin of this energy shift can be possibly traced to either noise and errors in real processors, as indicated by our previous noise-containing simulations presented above in Figure 9, where a vast majority of the resulting energies were above the exact one, or alternatively, it may be that the correct solution was not found within a two-qubit space of our Hamiltonian.

## 3. Discussion

Our analysis above clearly showed that the simulations of runs of ideal quantum computers can be very sensitive to some aspects of the computational setup. In particular, it was interesting to see that all four inspected optimization methods provided sometimes (in at least 2% of the runs) the energy of excited states instead of the ground-state energy, for which we were primarily searching. While these results were scattered far away from the energy of the ground state between −1.2 Ha and −1.3 Ha (see Figure 2 and Figure 5) in the case of the simulations of the ideal quantum computer, the errors and noise related to real quantum processors would likely make the situation yet worse. The studied case of the energies of the H${}_{2}$ molecule is very convenient because the ground-state energy is separated from the other eigenvalues by quite a broad energy range of about 0.6 Ha. Consequently, it is relatively easy to distinguish the values related to the ground state from those related to the excited ones. However, other systems, for which the exact ground-state energy value will not be known a priori, may have the eigenvalues closer. It can become difficult to distinguish between computed energies that (i) are noise-related to the eigenvalue that we aimed at (such as the ground-state one) and (ii) those that are noise-related to other eigenvalues. We demonstrated how advantageous it is then to exploit the measured probabilities of different basis states. In particular, we analyzed the similarities between the sets of measured probabilities by employing either the Jaccard–Tanimoto similarity index or the scalar product (preceded by a square-root normalization of the measured probabilities).

We suggest that the similarity analysis become a topic of future studies, possibly employing other tools, such as cluster analysis, because of its advantages. First, a similarity analysis of measured probabilities can support the assessment of the computed energies, and our results indicated that it could distinguish cases that are just noisy from those that are related to other eigenvalues, such as higher-energy excited states. Second, the two similarity measures that we employed, the Jaccard–Tanimoto similarity index and the scalar product, performed clearly better when the ground-state energy was associated with one of the basis states having a probability close to one while the others had it close to zero. Third, various similarity measures have specific advantageous and disadvantageous characteristics. For example, when identifying states related to the ground state, the similarity analysis based on the scalar product seemed to be more successful than the one employing the J-T index. On the other hand, the former tended to overrate the similarity of the excited and completely erroneous states.

## 4. Methods

The electronic wave functions of the H${}_{2}$ molecule were searched in the Slater-type orbital basis set with each orbital expanded into 3 Gaussian functions (STO-3G). The distance between the atoms was set to 0.725 Å, and the Coulomb repulsion between nuclei was not taken into account. In order to use a quantum computer, it is necessary to transform the electronic Hamiltonian of the studied system from its first-quantization form into the second-quantization one [18]. This was further mapped onto qubit operators represented in the Pauli operator basis [42] by a suitable transformation, e.g., the Bravyi–Kitaev [43] (BK) or Jordan–Wigner [44] (JW) one. We used the former transformation employing the *qiskit* package [45]. Our code is available in [46]. The final 4-qubit Hamiltonian is then:(1)$$\begin{array}{cc} {\hat{H}}_{H_{2}}^{4-\mathrm{qubit}} & =c_{0}\;1+c_{1}\;Z_{0}+c_{2}\;Z_{1}Z_{0}+c_{1}\;Z_{2}+c_{2}\;Z_{3}Z_{2}Z_{1}+c_{3}\;Z_{1}+c_{4}\;Z_{2}Z_{0}\\ \; & +c_{5}\;X_{2}Z_{1}X_{0}+c_{6}\;Z_{3}X_{2}X_{0}+c_{6}\;X_{2}X_{0}+c_{5}\;Z_{3}X_{2}Z_{1}X_{0}\\ \; & +c_{7}\;Z_{3}Z_{2}Z_{1}Z_{0}+c_{7}\;Z_{2}Z_{1}Z_{0}+c_{8}\;Z_{3}Z_{2}Z_{0}+c_{3}\;Z_{3}Z_{1}. \end{array}$$

The *Z* and *X* terms represent Pauli operators, and the coefficients $c_{i}$ are integrals evaluated by the *qiskit.chemistry* package [47]: $c_{0}$ = −0.80718, $c_{1}$ = 0.17374, $c_{2}$ =−0.23047, $c_{3}$ = 0.12149, $c_{4}$ = 0.16940, $c_{5}$ = −0.04509, $c_{6}$ = 0.04509, $c_{7}$ = 0.16658, $c_{8}$ = 0.17511. To solve this Hamiltonian, we used one of the most popular quantum-computing approaches, the variational quantum eigensolver (VQE) [45,48,49], involving a quantum circuit, as depicted in Figure 3. Referring to the VQE description, such as in, e.g., the recent review [49], the VQE starts by initializing the qubit register (the left-most part of Figure 3).

A quantum circuit is then applied to the qubit register in order to model the physics and entanglement of the electronic wavefunctions. A quantum circuit consists of a series of quantum operations on the qubits [7]. The structure of the circuit, i.e., a set of ordered quantum gates, is called an “ansatz”, and its behavior is defined by a set of parameters (the middle part of Figure 3). Once qubits are initialized, their state is designed to model a trial wavefunction. The Hamiltonian of the studied system can be measured with respect to this wavefunction to estimate the energy (the right-most part of Figure 3). As the measurement of the expectation value is performed only in the computational (*Z*-)basis, the terms in our qubit Hamiltonian containing the Pauli *X* operators must be measured in a non-diagonal basis. Therefore, basis-switching single-qubit gates (post-rotations) were utilized; see Figure 3.

The VQE then variationally optimizes the parameters of the ansatz in order to minimize the trial energy [50,51,52]. This is a hybrid approach when a classical computer is used to determine a new set of state-defining parameters based on the measurements of the quantum circuit employing a classical optimization method. In other words, the optimization was performed in a classical manner, but the determination of the cost-function, the energy of the state in our case, was outsourced to a quantum machine. This allowed the use of quantum computers that are readily available, but are rather small and noisy.

Not all of the Pauli terms in our Hamiltonian need to be determined individually as the Pauli operators that require the same post-rotations in the tensor product basis sets can be grouped. Only two circuits (which we will call Circuits 0 and 1 below) were needed in our case. Numerous runs (so-called shots) and measurements of the probabilities of the basis states were needed to obtain reliable expectation values for these two circuits (we used 4096 shots, unless specified otherwise) [53].

We mostly used simulations of quantum-computer runs, both ideal without noise and noisy ones, but our simulations were complemented also by experiments using real quantum devices. To access them, we used the publicly available cloud-based quantum computing platform IBM Quantum [54]. In order to save these precious resources, we used only the 2-qubit Hamiltonian. This was possible because the Hamiltonian in Equation (Equation 1) commutes with $Z_{1}$ and with $Z_{3}$. Consequently, the Hamiltonian is block-diagonal with 4 blocks when each of them corresponds to a particular computational basis setting of Qubits 1 and 3. This opens the way toward finding a Hamiltonian with the same ground-state energy expressed in a 2-qubit space [47]:(2)$${\hat{H}}_{H_{2}}^{2-\mathrm{qubit}}=c_{0}\;1+c_{1}\;Z_{0}+c_{1}\;Z_{1}+c_{2}\;Z_{1}Z_{0}+c_{3}\;X_{1}X_{0},$$
with the coefficients equal to $c_{0}$ = −1.05016, $c_{1}$ = 0.40421, $c_{2}$ = 0.01135, and $c_{3}$ = 0.18038.

## 5. Conclusions

Motivated by the progress of quantum computing in quantum chemistry, we performed a series of simulations of quantum-computer runs to inspect the practical aspects of such calculations. We calculated the ground-state energy of the hydrogen molecule as a benchmark example using the variational quantum eigensolver (VQE). We focused on the impact of the choice of the optimization method (COBYLA, SPSA, Nelder–Mead, and Powell’s), as well as the architecture of the quantum circuits and different types of noise (gate errors and readout errors).

The SPSA and COBYLA methods significantly outperformed the Nelder–Mead and Powell optimization methods. The results obtained when using the $R_{y}$ variational form were found better than those when the $R_{y}R_{z}$ form was used. In particular, the statistical distribution of computed energies spanned a narrower range and was closer to the exact value. Further, the iterative runs were less likely to lead to excited states instead of the ground state, at which that we primarily aimed. The choice of an optimum entangling layer was interlinked with the choice of the optimization method. For example, the performance of the COBYLA method was found to worsen when the circular entangling layer was used, while the full-entangling layer improved its performance. In contrast to this sensitivity of the COBYLA method, the SPSA method turned out to be quite robust with respect to the entangling layer(s) type.

Further, all four inspected optimization methods sometimes led to an energy that corresponded to an excited state rather than the ground state. We showed that a similarity analysis of the measured probabilities using the Jaccard–Tanimoto similarity index or the scalar product could provide a very useful insight into these cases. The similarity analysis of the measured probabilities could support the assessment of the computed energies, and our results indicated that it could distinguish cases that are just noisy from those that are related to other eigenvalues, such as excited states with higher energies.

Importantly, both similarity measures performed clearly better when the ground-state energy was associated with the situation when one of the basis states had a probability equal to one while the others had at zero. In contrast to the above-mentioned characteristics, which are common to both similarity measures used, there are significant differences between them as well. In particular, when identifying states related to the ground state, the similarity analysis based on the scalar product seemed to be more successful than that employing the J-T index. On the other hand, the former tended to overrate the similarity of the excited and completely erroneous states. 

## Figures and Tables

**Figure 1 molecules-27-00597-f001:**
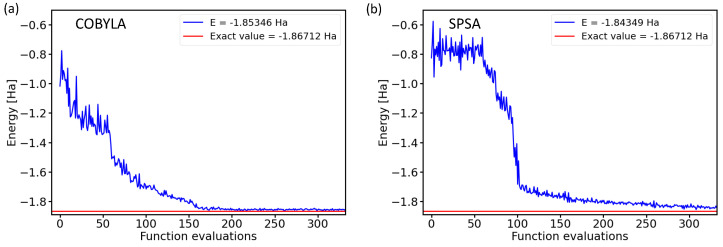
Dependencies of the ground-state energies of the H${}_{2}$ molecule as a function of the number of iterations during the optimization process by the classical optimization methods COBYLA (**a**) and SPSA (**b**) in the case of a 4-qubit Hamiltonian obtained by the Bravyi–Kitaev transformation with the circuit architecture characterized by the $R_{y}R_{z}$ variational form and the linear entangling layer of qubits.

**Figure 2 molecules-27-00597-f002:**
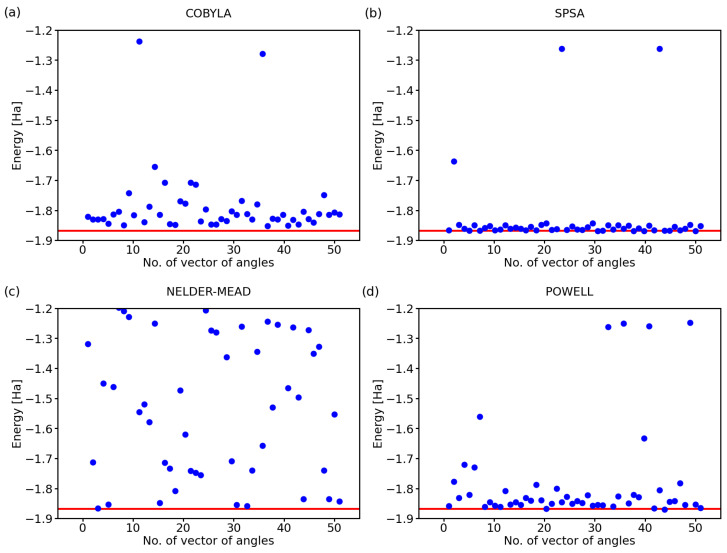
Computed ground-state energies of the H${}_{2}$ molecule when using 50 different initial sets of random angles and four classical optimization methods: COBYLA (**a**), SPSA (**b**), Nelder–Mead (**c**), and Powell (**d**). The employed 4-qubit Hamiltonian was obtained by the Bravyi–Kitaev transformation; the circuit exhibited the $R_{y}R_{z}$ variational form, linear entangling layer, and the evaluations were based on 4096 shots. The red line represents the exact value.

**Figure 3 molecules-27-00597-f003:**
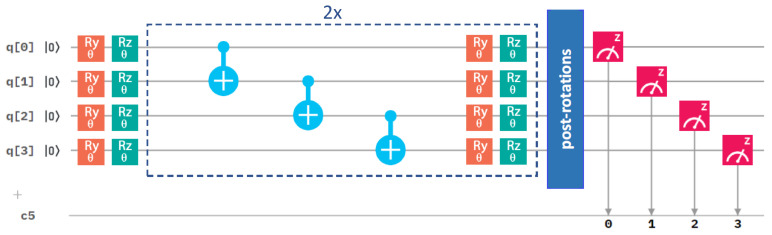
A schematic of the quantum circuit used with the $R_{y}R_{z}$ variational form and the linear entangling layer of qubits. It contains gates rotating the qubits (orange and green symbols), control NOT gates responsible for entangling layers (light blue symbols), and measurement (red symbol). The central part marked by blue dashed lines is performed twice sequentially in our circuit. c5 denotes a classical register of five bits where the results of the measurements are recorded. One of the bits is not used, as we only make measurement on four qubits.

**Figure 4 molecules-27-00597-f004:**
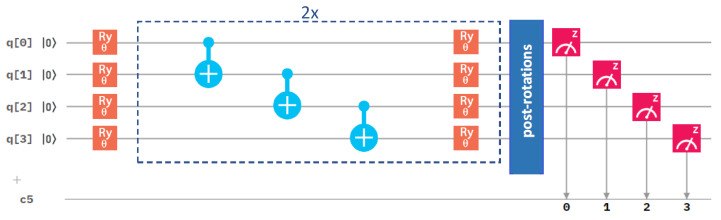
A schematic visualization of a circuit containing the $R_{y}$ variational form and the linear entangling layer of qubits. The central part marked by blue dashed lines was sequentially repeated twice in our circuit.

**Figure 5 molecules-27-00597-f005:**
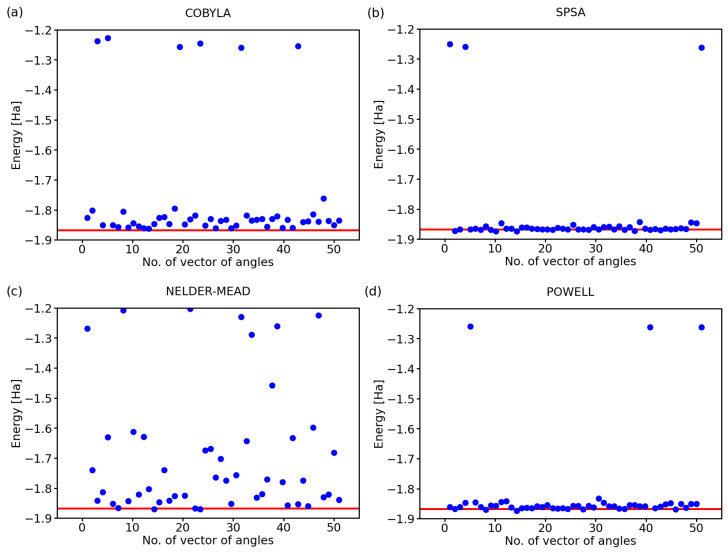
Similar sets of results as in Figure 2, i.e., obtained using classical optimization methods, COBYLA (**a**), SPSA (**b**), Nelder–Mead (**c**), and Powell (**d**), but the variational form as $R_{y}$ (other circuit characteristics are the same as in the case of Figure 2).

**Figure 6 molecules-27-00597-f006:**
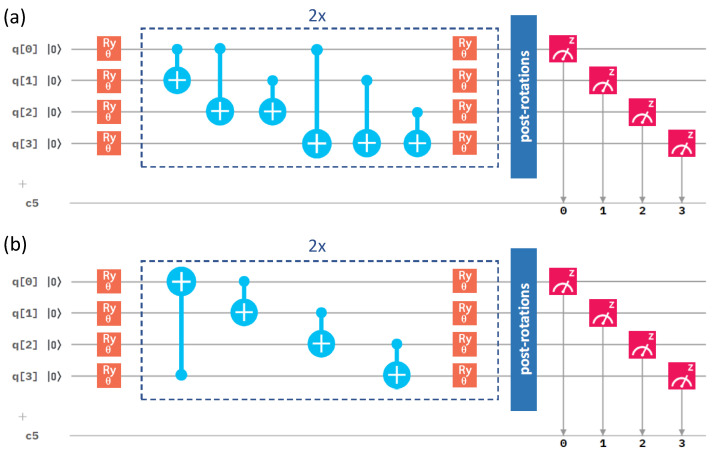
Two additional types of qubit entangling layers that we compared with the linear one. In particular, we show a full-entangling layer (**a**) and the circular one (**b**).

**Figure 7 molecules-27-00597-f007:**
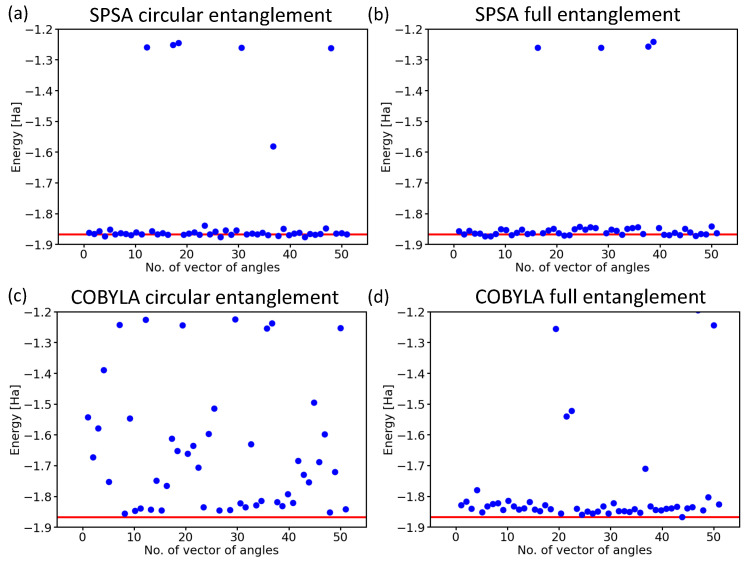
Computed ground-state energies of the H${}_{2}$ molecule as obtained for 50 different initial sets of random angles in the case of the SPSA optimization method with the circular entangling layer (**a**) and the full one (**b**), as well as the COBYLA optimization method with either the circular entangling layer (**c**) or the full one (**d**). Results are for the $R_{y}$ variational form and 4096 shots.

**Figure 8 molecules-27-00597-f008:**
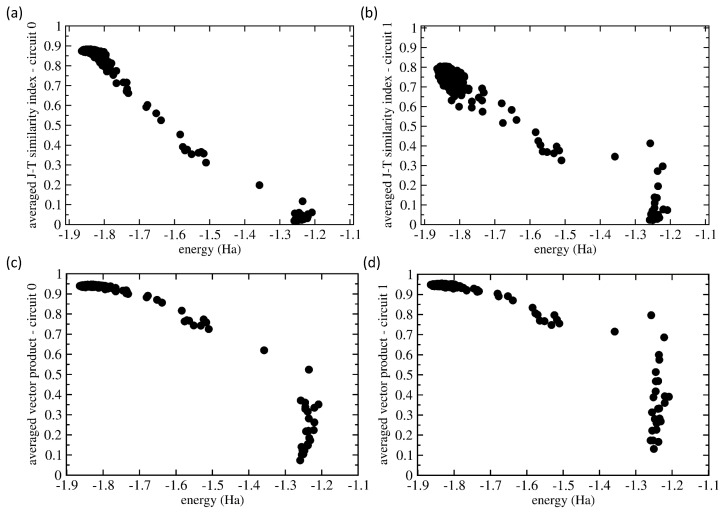
Comparison of the similarities of the vectors of the measured probabilities as functions of the energy. The plotted values are averaged ones using all 500 values and evaluated using either the Jaccard–Tanimoto (J-T) similarity index (**a**,**b**) and scalar product (**c**,**d**) for both Circuit 0 (**a**,**c**) and Circuit 1 (**b**,**d**).

**Figure 9 molecules-27-00597-f009:**
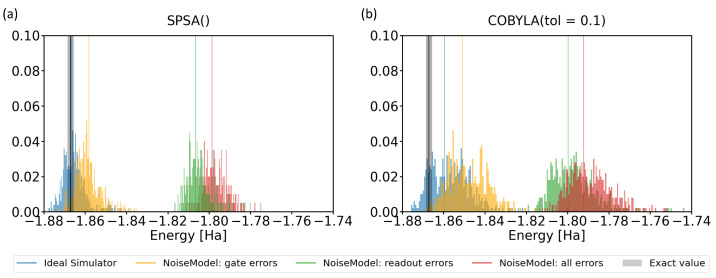
Results of our simulations when various types of noise were included using either the SPSA (**a**) or the COBYLA (**b**) optimization method. The simulations aimed at determining the ground-state energy of the H${}_{2}$ molecule in the case of the 4-qubit Hamiltonian. The grey vertical thick line represents the energy region around the exact result corresponding to the chemical accuracy. Thin vertical lines of different colors indicate the median of various sets of results. The minimizations by the SPSA method used default parameters, while those by the COBYLA method were characterized by a tolerance value equal to 0.1.

**Figure 10 molecules-27-00597-f010:**
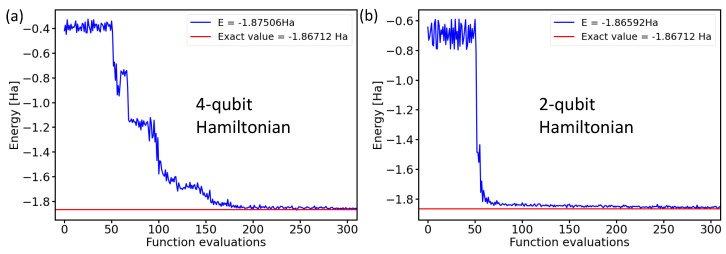
Comparison of the calculated ground-state energies of the H${}_{2}$ molecule in the case of the 4-qubit Hamiltonian (**a**) or the 2-qubit Hamiltonian (**b**) using the $R_{y}$ variational form, the linear entangling layer, the SPSA optimization method, and the probabilities based on 4096 shots.

**Figure 11 molecules-27-00597-f011:**
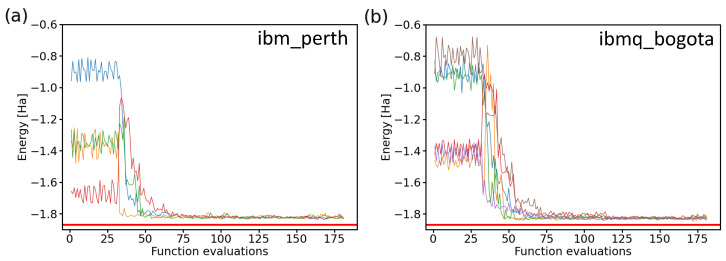
Comparison of 10 iterative runs (color-coded) aimed at determining the ground-state energies of the H${}_{2}$ molecule using two IBM quantum processors. We show four runs in the case of the ibm_perth quantum processor (**a**) and the six ibmq_bogota runs (**b**) as a function of the number of function evaluations. The results were obtained for a 2-qubit Hamiltonian, the $R_{y}$ variational form, the linear entangling layer of qubits, the SPSA optimization method (the maxiter parameter equal to 75), and the probabilities based on 8192 shots.

## Data Availability

The data presented in this study are available upon request from the corresponding author.

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
