# Peer review of "Best-Practice Aspects of Quantum-Computer Calculations: A Case Study of the Hydrogen Molecule"

_molecules, 2022, doi:10.3390/molecules27030597_

Round 1

Reviewer 1 Report

The authors study the performance of the simulations with different quantum computational optimization methods, different quantum circuits, and different noises. They show the better calculation efficiency and accuracy of COBYLA and SPSA optimization methods compared to the other traditional methods when applying them on calculating the ground state energy of hydrogen molecules. Furthermore, the variation form and basis states play vital roles in the calculation of different methods, which makes sense. Although the authors designed some simple experiments to test, the results are very clear and efficient. I recommend publishing this paper on Molecules with minor revisions.

Some concerns:

  1. In Fig2.(b), is the x-axis label wrong?
  2. How much time do you cost on the IBM Quantum? Will it be quite different from other public quantum computers?
  3. The Conclusion paragraph rather needs to be rewritten. It is almost the same as the abstract.

Author Response

  1. Iterations and function evaluations have basically the same meaning here, but we have changed the label of Fig. 2a to keep coherence between both graphs as well as with Fig. 10 and Fig. 11.
  2. We have used IBM quantum computer as it allows a rather generous access to different processors without fees. Most calculations were performed using the public queue, some using the elevated researchers’ account allowing dedicated use of some of the processors for two hours a month. Most of the results presented in the manuscript are based on simulations performed on desktop PC computer, the total time of simulations has been estimated to a few hundred hours.
  3. We have significantly rewritten the conclusion paragraph as suggested.

Reviewer 2 Report

Review work entitled “Best-practice aspects of quantum-computer calculations: A case study of hydrogen molecule” written by Mihalikova et al. presents pioneer study employing quantum computer resources to examine best-practice aspects of such calculations.

It is worth emphasizing that the Authors made excellent designing choice by selecting very simple hydrogen molecule as testing object. Such approach allowed to show these complexed and modern computational models on the basis of a fundamental entity.

This work is undoubtedly significant one as quantum computers are the future of molecular modelling and probably the best possible and most accurate tool in theoretical chemistry.

I consider this work as vital, as it brings fresh insights into the area of theoretical prediction of molecules geometries and properties.

Therefore, in my opinion this work deserves to be published in Molecules as it is.

Author Response

We very much appreciate such a favorable opinion.

Reviewer 3 Report

Achieving accurate simulations of molecular systems or condensed matter would lead to breakthroughs in fundamental science and applications, such as medicine and materials science. However, this is computationally intractable even for the best supercomputers. At current, with the readiness of control of dozens of qubits, the community is investigating useful quantum simulation ran on existing and near-term, noisy intermediate-scale quantum (NISQ) processors.  Along this line, the IBMQ is developing one of the leading quantum computing platforms and a comprehensive “quantum” ecosystem for researchers to access.

This article used H2 as a case study of the well-known VQE (variational quantum eigensolver) and benchmarked several optimization methods under various setups and noisy configurations. This work studies the impact of several computational setups in the VQE ground state searching and covers implementation experimentally on the IBM quantum devices. Overall, the scientific content is fine; the article is written clearly, and the figures are delivered professionally. I suggest its publication and I have several minor observations to address as follows. 

  1. In Section 2, through lines 56-69, the VQE is introduced. Considering the journal -Molecules- is not specifically quantum-based, the introduction can be improved by explaining how VQE is a hybrid algorithm. The current version looks not specify that the quantum part is just to “evaluate” the ground state through measurement statistics based on a trail configuration, and the classical part is used to update the new trial and feed it to the quantum device.
  2. Figure-1: it’s worth noting what is the “c5” where.
  3. At the beginning of Section 3, the simulation of COBYLA and SPSA are compared.? In addition to the convergence track and result precision, what are their running speed? Are they on the same time scale, as this is not specified otherwise in the paper
  4. Section 3.1, at line 99, the “randomly selected initial set of angles.” Is this a random sampling on the Bloch sphere by two random variables?
  5. To the best of my knowledge, the performance of VQE typically relies on wavefunction ansatz that results in approximate wavefunctions and energies. Iff the initial state is spin-up (given by a random seed) while the true ground state is spin-down, such a “bad” trial state will probably deteriorate the VQE’s performance. In view of this, the performance of the four methods is only displayed and compared, while a discussion paragraph is absent and will lower the impact of the paper. [I am not expecting a strict analysis]
  6. Out of curiosity, in Fig.5, why is the layout named “circular” one?
  7. In Line 149, a typo: Fig.6 should be Fig.7.
  8. In Figure.9, the “thick” and the “thin” legend are not consistent with the caption.

Author Response

  1. We have extended the description of the VQA algorithm to include the explanation as suggested.
  2. We have included the description of the classical register.
  3. The bottleneck of both simulations and runs on real quantum machines is the function evaluation, i.e. the estimation of the energy of the probe state. The calculation of a new state, be it by SPSA or Cobyla, has only a negligible contribution to the running time. Thus Iterations, or function evaluations directly correspond to the running time. Translation to physical time dimension like seconds however strongly depends on the infrastructure used, for simulations the power of the classical computer and for real runs the speed of the quantum device, queue load, calibration breaks etc.
  4. As we are using two or four qubits, the sampling has to cover a much broader space than just a Bloch sphere of a single qubit. The coverage also depends on the type and number of entangling layers used. The random selected set thus has to be related rather to the set of initial parameters (preparation-type dependent) than to randomly covering the whole set of states.
  5. We believe there is a slight misunderstanding in this point. In our context, the wavefunction ansatz is represented by the type and number of entangling layers used. In our example, all the entangling layer configurations were sufficient to cover the ground state, thus it was only a possible ignorance of the minimization function if the correct result was not found. For more complicated situations, which however are not covered by our manuscript, it might happen that a specific configuration (e.g. a single entangling layer) would not cover the ground state, leading to impossibility to reach it using any minimization method. As we have focused on the performance of different minimization functions rather than of the analysis of optimal preparation procedures, we believe it would be misleading to discuss this topic within the manuscript in more detail.
  6. Probably Fig. 6 was meant here. Circular comes from the fact that the C-Not gates are connected in a circle, 1-2, 2-3, 3-4, 4-1.
  7. We have corrected the typo.
  8. We have changed the caption and the color of the chemical accuracy region to grey to avoid any confusion.